# Hybrid Combinatorial Multi-armed Bandits with Probabilistically Triggered Arms

## Abstract

The problem of combinatorial multi-armed bandits with probabilistically triggered arms (CMAB-T) has been extensively studied. Prior work primarily focuses on either the online setting where an agent learns about the unknown environment through iterative interactions, or the offline setting where a policy is learned solely from logged data. However, each of these paradigms has inherent limitations: online algorithms suffer from high interaction costs and slow adaptation, while offline methods are constrained by dataset quality and lack of exploration capabilities. To address these complementary weaknesses, we propose hybrid CMAB-T, a new framework that integrates offline data with online interaction in a principled manner. Our proposed hybrid CUCB algorithm leverages offline data to guide exploration and accelerate convergence, while strategically incorporating online interactions to mitigate the insufficient coverage or distributional bias of the offline dataset. We provide theoretical guarantees on the algorithm's regret, demonstrating that hybrid CUCB significantly outperforms purely online approaches when high-quality offline data is available, and effectively corrects the bias inherent in offline-only methods when the data is limited or misaligned. Empirical results further demonstrate the consistent advantage of our algorithm.

## 1 Introduction

Combinatorial multi-armed bandits with probabilistically triggered arms (CMAB-T) provide a powerful framework for modeling a broad class of real-world sequential decision-making problems, including influence maximization, learning to rank, and large language model cache (Chen et al., 2013; 2016; Wang & Chen, 2017; Wen et al., 2017; Kong et al., 2023; Liu et al., 2023; 2025; Pope et al., 2022; Zhu et al., 2023; Gim et al., 2023; Qu et al., 2024). In these settings, a decision-maker repeatedly selects a combinatorial action, typically a subset of base arms, and receives partial feedback governed by a probabilistic triggering process.

Most existing work on CMAB-T has focused on the *online setting*, where an agent learns through trial and error by interacting with the environment over multiple rounds (Chen et al., 2013; 2016; Wang & Chen, 2017; Wen et al., 2017; Kong et al., 2023; Liu et al., 2023; 2024). While this approach enables adaptive learning and active exploration, it often incurs high feedback collection costs and suffers from slow convergence—particularly in large-scale or high-stakes domains.

A study (Liu et al., 2025) has begun to explore the *offline setting* for CMAB-T, where the goal is to learn decision policies from pre-collected data logs, thereby avoiding the expense of online interaction. However, offline learning is highly sensitive to the quality and coverage of the logged data. For example, rare but important action combinations may be missing, and distributional shifts between the offline data set and online environment can lead to suboptimal performance. Moreover, the lack of active exploration limits the learner's ability to gather information about underexplored or high-uncertainty actions.

The limitations of purely online or offline learning motivate the study of hybrid learning methods, which use offline data to warm-start online learning (Shivaswamy & Joachims, 2012; Song et al., 2023; Oetomo et al., 2023; Agnihotri et al., 2024; Cheung & Lyu, 2024; Qu et al., 2025). These approaches balance the cost-free nature of offline data with the adaptability of online exploration, often leading to improved sample efficiency in practice. While hybrid methods have been studied

in the classical MAB problems, their extension to the general CMAB-T setting remains largely unexplored.

The technical challenge arises when incorporating the offline data into the regret analysis of the online CMAB-T. In particular, we must determine *when* to rely on the pure online observation and *when* the offline data (may be biased) is sufficiently reliable to be used. In the MAB setting, the regret admits a clean decomposition: it can be expressed as the sum over arms of the number of times each suboptimal arm is pulled, multiplied by its corresponding sub-optimality gap. This makes it straightforward to quantify how offline data reduces regret by decreasing the selection count of suboptimal arms (Cheung & Lyu, 2024). But in our considered CMAB-T setting, such gap-based reasoning is no longer directly applicable, where the per-round regret cannot be attributed to individual arms through simple suboptimality gaps. The regret depends on the triggered arms and the combinatorial reward structure, making it much more difficult to define a universal threshold for determining when to use offline data.

To overcome these challenges, our work focuses on the following fundamental questions:

*(1) How to derive an algorithm that effectively leverages offline data in the online CMAB-T setting?*

*(2) Can we provide the corresponding theoretical guarantees that offline data leads to measurable improvement compared with purely online algorithms?*

We answer these questions through the following contributions:

**Problem Formulation.** We formally define the *hybrid CMAB-T* (H-CMAB-T) setting by extending the classical CMAB-T framework to incorporate offline data. In particular, we define the offline dataset as a collection of observations over base arms, and introduce a notion of *bias* based on the discrepancy between the offline and online mean rewards of each arm. This formulation provides a principled basis for assessing when offline data can be beneficial to online learning.

**Algorithm Design.** We propose a new algorithm *hybrid CUCB* leveraging the biased offline data to improve the classic CUCB algorithm. This algorithm balances offline and online feedback through a dual-UCB mechanism. Specifically, we construct two confidence bounds for each base arm: one purely based on the feedback collected online, and another that hybridizes observations from both the offline data set and online interactions with an explicit bias correction. By selecting the minimum of the two UCB estimates, the algorithm adaptively leverages the offline data based on its quality.

**Theoretical Analysis.** To overcome challenge from the core difference between MAB and CMAB-T, we draw on the intuition that while the bias may appear at the level of individual arms, the regret in CMAB-T arises from actions that involve multiple arms and triggering mechanisms. Motivated by this, we explore a connection between per-arm bias and action-level regret by considering a hypothetical allocation of the regret to the arms that could be triggered in each round. This perspective allows us to bridge the arm-level discrepancy introduced by offline data and the combinatorial nature of regret in CMAB-T. Leveraging this connection, we construct a threshold condition that determines whether the offline estimates are reliable enough to be used. Finally, We provide both gap-dependent and gap-independent regret bounds. Our results show that the algorithm achieves improved regret over standard online methods (Wang & Chen, 2017), with a provable *saving term* that depends on the informativeness and reliability of the offline data. Our result recovers the standard online regret when offline data is absent or adversarial, and it matches or improves upon the results of Cheung & Lyu (2024) when the problem reduces to classical MAB.

**Empirical Evaluation** We complement our theoretical analysis with empirical evaluations. The results consistently demonstrate that hybrid CUCB outperforms both purely online and purely offline baselines, highlighting its adaptability and robustness across varying data conditions.

## 2 RELATED WORK

**Online Bandits.** MAB problems have been extensively studied as a foundational model for sequential decision-making under uncertainty (Auer et al., 2002; Bubeck & Cesa-Bianchi, 2012; Lattimore & Szepesvári, 2020). The combinatorial multi-armed bandit (CMAB) framework (Chen et al., 2013) generalizes classical MAB by allowing the learner to select subsets of arms (super arms) in each round, leading to richer modeling power and broader applicability. In particular, the CMAB with

probabilistically triggered arms (CMAB-T) framework introduced by Chen et al. (2016); Wang & Chen (2017) captures the settings such as influence maximization, online learning to rank where the reward depends not only on the chosen super arm but also on a random triggering process. This framework has also been extended to incorporate contextual information (Liu et al., 2023). A line of work has established algorithms with theoretical regret guarantees under structural assumptions such as monotonicity and bounded smoothness (Chen et al., 2016; Wang & Chen, 2017; Wen et al., 2017; Liu et al., 2022; 2023; 2024). All these approaches operate entirely in the online setting.

**Offline Bandits.** Offline learning in bandit and reinforcement learning has gained increasing attention due to the high cost of online exploration and the availability of logged historical data. It has been explored in many bandits settings like the classical MAB (Rashidinejad et al., 2021), contextual MAB (Rashidinejad et al., 2021; Jin et al., 2021; Li et al., 2022) and neural contextual bandits (Nguyen-Tang et al., 2021; 2022). For combinatorial bandits, Liu et al. (2025) recently propose CLCB, the first general framework for offline learning in CMAB problems, which characterizes dataset quality through coverage conditions, and provide near-optimal theoretical guarantees.

**Hybrid Bandits.** To mitigate the limitations of purely online or offline learning, hybrid methods aim to combine their respective advantages by using offline data to initialize or guide online exploration. Hybrid learning has been studied in various domains, including bandit problems (Shivaswamy & Joachims, 2012; Oetomo et al., 2023; Agnihotri et al., 2024) and reinforcement learning (Song et al., 2023; Qu et al., 2025). Most of these hybrid methods assume that offline data is unbiased and directly compatible with the online environment (Shivaswamy & Joachims, 2012; Song et al., 2023; Oetomo et al., 2023; Agnihotri et al., 2024). Qu et al. (2025) assume a strongly biased offline dataset with a lower bound on the discrepancy between offline and online means. Cheung & Lyu (2024) do not require such assumptions and propose an algorithm that adaptively incorporates offline data based on its reliability. To the best of our knowledge, the hybrid learning problem in CMAB-T remains open.

## 3 PROBLEM SETUP

We first introduce the *hybrid combinatorial mutlti-armed bandits with probabilistically triggered arms* (H-CMAB-T) problem. The H-CMAB-T problem explored in this paper is built upon the standard CMAB-T framework (Wang & Chen, 2017). We begin by reviewing the classical CMAB-T setting, and then introduce how offline data is incorporated in our extension.

The online environment consists of $m$ base arms, represented as random variables $X_1, X_2, \ldots, X_m$, jointly distributed according to an unknown distribution $D^{\mathrm{on}} \in \mathcal{D}$, where $D^{\mathrm{on}}$ is supported on $[0,1]^m$ and $\mathcal{D}$ is the distribution family. For each base arm $i \in [m]$, let $\mu_i^{\mathrm{on}} = \mathbb{E}_{X \sim D^{\mathrm{on}}}[X_i]$ denote its expected value, and define the vector $\mu^{\mathrm{on}} = (\mu_1^{\mathrm{on}}, \ldots, \mu_m^{\mathrm{on}}) \in [0,1]^m$ as the mean vector of all arms. Note that $\mu^{\mathrm{on}}$ is determined by the underlying distribution $D^{\mathrm{on}}$. The learning process unfolds over discrete rounds $t = 1, 2, \ldots, T$. In each round:

*1.* The learner selects a combinatorial action $S_t \in \mathcal{S}$ based on the previous rounds observation and feedback, where $\mathcal{S}$ is a predefined action space, possibly subject to structural constraints. The combinatorial action $S_t$ is also called "super arm" and in many cases it is a subset of base arms.

*2.* The environment draws an independent sample $X^{(t)} = (X_1^{(t)}, \ldots, X_m^{(t)}) \sim D^{\mathrm{on}}$.

*3.* Playing action $S_t$ in the environment induces a random subset $\tau_t \subseteq [m]$ of arms to be triggered. The triggering process is stochastic: even given the environment outcome $X^{(t)}$ and the chosen action $S_t$, the triggered set $\tau_t \subseteq [m]$ may still exhibit randomness. We model this using a *probability triggering function* $D^{\mathrm{trig}}(S, X)$, which defines a distribution over subsets of $[m]$ conditioned on action $S$ and environment realization $X$. Formally, we assume that for each round $t$, the triggered set $\tau_t$ is independently drawn from $D^{\mathrm{trig}}(S_t, X^{(t)})$, i.e., $\tau_t \sim D^{\mathrm{trig}}(S_t, X^{(t)})$. Moreover, to enable algorithms to estimate $\mu_i^{\mathrm{on}}$ from observed samples during online learning, we make the following identifiability assumption: the outcome of each arm $i$ does not depend on whether it is triggered. That is, $\mathbb{E}_{X \sim D^{\mathrm{on}}, \tau \sim D^{\mathrm{trig}}(S, X)}[X_i \mid i \in \tau] = \mathbb{E}_{X \sim D^{\mathrm{on}}}[X_i] = \mu_i^{\mathrm{on}}, \ \forall \, i \in [m]$.

*4.* A non-negative reward $R(S_t, X^{(t)}, \tau_t) \in \mathbb{R}_{\geq 0}$ is revealed to the learner, which is a deterministic function of the chosen action $S_t$, the sampled instance $X^{(t)}$, and the triggered set $\tau_t$. The expected

reward of an action $S \in \mathcal{S}$ is given by $r_S(\mu) := \mathbb{E}[R(S, X, \tau)]$, where the expectation is taken over $X \sim D$ and $\tau \sim D^{\mathrm{trig}}(S, X)$. We emphasize that $r_S(\mu)$ is a function of $S$ and the mean vector $\mu$.

The goal of the learner is to maximize the total expected reward over $T$ rounds, i.e., to design a learning algorithm that selects $S_1, \ldots, S_T$ to maximize $\sum_{t=1}^{T} \mathbb{E}[R(S_t, X^{(t)}, \tau_t)]$.

While the classical CMAB-T framework captures the core structure of combinatorial bandit problems with triggering, it assumes that all learning happens online from scratch. In many practical scenarios, however, a significant amount of data is already available prior to online interaction—collected from historical logs or prior deployments. For example, in *online influence maximization* problem, the organizations often have access to past propagation traces—records of how information spread—which can serve as valuable offline data to accelerate online learning in new deployment scenarios.

Motivated by this, we consider an extension of CMAB-T that incorporates such *offline data*, and investigate how it can be used to improve learning performance. More specifically, the key difference between H-CMAB-T and CMAB-T problem is that before online learning, the player is given an offline data collection $\mathcal{B}$. It is worth noting that there may be discrepancies between offline data and the online environment. For example, in the OIM problem, due to the characteristics of the product or shifts in user preferences, the diffusion dynamics within social networks can differ. To characterize such phenomenon and avoid misleading of offline data, we consider that the arms in the offline data set and the online setting may have different means. Specifically, the outcomes of $m$ base arms in the offline data set can be represented as random variables $Y_1, Y_2, \ldots, Y_m$, jointly distributed according to an unknown distribution $D^{\mathrm{off}}$ and the mean vector of the offline data is $\mu^{\mathrm{off}} = (\mu_1^{\mathrm{off}}, \ldots, \mu_m^{\mathrm{off}})$. It is natural that $|\mu_i^{\mathrm{on}} - \mu_i^{\mathrm{off}}| \geq 0$, and equality holds if and only if the offline data is unbiased. Without loss of generality, we denote $N_i$ as the number of the independent observations of arm $i$. Then the offline data set can be represented as $\mathcal{B} := \{N_i, \{Y_{i,s}\}_{s=1}^{N_i}\}_{i=1}^{m}$.

**Bias control.** Besides, to quantify this discrepancy, we adopt the bias control vector $V = (V_1, \ldots, V_m)$ as a hyper-parameter which upper bounds the difference between the offline and online means for each arm:
$$|\mu_i^{\mathrm{off}} - \mu_i^{\mathrm{on}}| \leq V_i, \quad \forall i \in [m].$$
Since both means lie in $[0, 1]$, we assume $V_i \in [0, 1]$ for all $i$. Smaller values of $V_i$ indicate higher alignment between offline and online environments. In settings with prior knowledge—e.g., similar user populations or stable network dynamics—we may set $V_i$ to be small. In fully agnostic cases where no such knowledge is available, we conservatively set $V_i = 1$.

*Remark 1. As rigorously shown in Section 3 of Cheung & Lyu (2024), in the presence of biased offline data, no hybrid algorithm in MAB can be guaranteed to outperform a purely online baseline unless some prior knowledge about the bias is available. This theorem highlights that incorporating some form of prior understanding of the bias is not just helpful but fundamentally necessary. To understand this challenge, one can consider the unknown $V$ setting and try to design a hybrid algorithm that learns $V$ during the online interaction. This raises a challenging trade-off: if $V$ is small, estimating it accurately may require excessive online samples, outweighing the benefit of offline data; if $V$ is large, offline estimates are often too biased to be useful, making a pure online strategy preferable. Exploring the unknown $V$ setting is valuable but technically demanding, and we leave it as an important direction for future work.*

Consequently, based on the above problem formulation, we define an H-CMAB-T instance as a tuple $([m], \mathcal{S}, \mathcal{D}, D^{\mathrm{trig}}, R, \mathcal{B})$. To make the learning problem well-defined and practically solvable, it remains to specify how actions are selected given current estimates of the arm statistics. In many CMAB-T instances, the action space is exponentially large and the underlying optimization problem of selecting the optimal super arm is NP-hard (Chen et al., 2013; 2016). To decouple the statistical estimation from the combinatorial optimization, prior works commonly assume the access to an *offline oracle* that returns an approximate solution. This allows the learning algorithm to focus on estimating arm statistics while relying on the oracle to select actions.

**Offline $(\alpha, \beta)$-approximation oracle $\mathcal{O}$.** We assume access to an offline $(\alpha, \beta)$-approximation oracle, denoted by $\mathcal{O}$. This oracle takes as input the mean vector $\mu = (\mu_1, \ldots, \mu_m)$ and returns an action $S^{\mathcal{O}} \in \mathcal{S}$ such that $\mathbb{P}\left[r_{S^{\mathcal{O}}}(\mu) \geq \alpha \cdot \mathrm{opt}_\mu\right] \geq \beta$, where $\alpha \in (0, 1]$ is the approximation ratio, and $\beta \in (0, 1]$ is the success probability. Here, $\mathrm{opt}_\mu$ denotes the optimal expected reward under mean vector $\mu$, defined as $\mathrm{opt}_\mu := \sup_{S \in \mathcal{S}} r_S(\mu)$. And we assume that $\mathrm{opt}_\mu$ is bounded for all $\mu$.

Further, the objective of the learner is to minimize the $(\alpha, \beta)$–approximation regret defined as below (Chen et al., 2013; 2016; Wang & Chen, 2017; Wen et al., 2017).

**Definition 1** $((\alpha, \beta)$-approximation regret.**)**. *The $(\alpha, \beta)$-approximation regret of a learning algorithm $\mathcal{A}$ over $T$ rounds under an H-CMAB-T instance $([m], \mathcal{S}, \mathcal{D}, D^{trig}, R, \mathcal{B})$ is*

$$\mathrm{Reg}^{\mathcal{A}}_{\mu^{on}, \alpha, \beta}(T) := \alpha \cdot \beta \cdot T \cdot \mathrm{opt}_{\mu^{on}} - \mathbb{E}\left[\sum_{t=1}^{T} R(S_t^{\mathcal{A}}, X^{(t)}, \tau_t)\right] = \alpha \cdot \beta \cdot T \cdot \mathrm{opt}_{\mu^{on}} - \mathbb{E}\left[\sum_{t=1}^{T} r_{S_t^{\mathcal{A}}}(\mu^{on})\right],$$

*where $S_t^{\mathcal{A}}$ is the action selected by algorithm $\mathcal{A}$ at round $t$, and the expectation is taken over the randomness of the environment outcomes $\{X^{(t)}\}_{t=1}^{T}$, the triggered sets $\{\tau_t\}_{t=1}^{T}$, and the internal randomness of the algorithm.*

This notion of regret captures how far the cumulative reward falls short of what could be obtained by always playing a near-optimal action provided by the oracle.

We now introduce several conditions that are used to establish regret guarantees. These conditions are widely adopted in the CMAB literature (Chen et al., 2016; Wang & Chen, 2017; Wen et al., 2017; Liu et al., 2023; 2025). To facilitate the presentation, we denote $p_i^{D,S}$ as the probability that arm $i$ is triggered when action $S$ is selected in environment $D$.

**Condition 1** (Monotonicity). *We say that a CMAB-T problem instance satisfies monotonicity, if for any action $S \in \mathcal{S}$, for any two distributions $D, D' \in \mathcal{D}$ with expectation vectors $\boldsymbol{\mu} = (\mu_1, \ldots, \mu_m)$ and $\boldsymbol{\mu}' = (\mu_1', \ldots, \mu_m')$, we have $r_S(\boldsymbol{\mu}) \leq r_S(\boldsymbol{\mu}')$ if $\mu_i \leq \mu_i'$ for all $i \in [m]$.*

**Condition 2** (1-Norm TPM Bounded Smoothness). *We say that a CMAB-T problem instance satisfies 1-norm TPM bounded smoothness, if there exists $B \in \mathbb{R}^+$ (referred as the bounded smoothness constant) such that, for any two distributions $D, D' \in \mathcal{D}$ with expectation vectors $\boldsymbol{\mu}$ and $\boldsymbol{\mu}'$, and any action $S$, we have $|r_S(\boldsymbol{\mu}) - r_S(\boldsymbol{\mu}')| \leq B \sum_{i \in [m]} p_i^{D,S} |\mu_i - \mu_i'|$.*

The two reward function conditions encode natural intuitions in the CMAB-T setting: Condition 1 reflects monotonicity—if all arm means are higher in one set than another, any action should yield a higher expected reward; Condition 2 captures the role of triggering probabilities—arms that are triggered more often contribute more to the reward and thus require more accurate mean estimates, while less frequently triggered arms can tolerate greater uncertainty.

# 4 THE HYBRID CUCB ALGORITHM

In this section, we provide an algorithm, hybrid CUCB (Algorithm 1), aiming to leverage *useful* offline data to accelerate the online learning efficiency. The hybrid CUCB algorithm runs as follows. In each round, the algorithm computes two UCB vectors:

$$\mathrm{UCB}_t = (\mathrm{UCB}_t(1), \ldots, \mathrm{UCB}_t(m)), \quad \mathrm{UCB}_t^{\mathrm{S}} = (\mathrm{UCB}_t^{\mathrm{S}}(1), \ldots, \mathrm{UCB}_t^{\mathrm{S}}(m)),$$

and then feeds the coordinate-wise minimum two of them into the $(\alpha, \beta)$-approximation oracle to select an action.

The vector $\mathrm{UCB}_t$ follows the standard CUCB construction (Wang & Chen, 2017) (Line 6 and 8), representing the conventional UCB established with the pure online feedback, where $T_i$ denotes the number of times that arm $i$ has been triggered.

As to H-CMAB-T problem, to effectively leverage offline data while remaining robust to distributional mismatch, we design a hybrid confidence bound $\mathrm{UCB}_t^{\mathrm{S}}$ that adaptively incorporates offline observations. Intuitively, when the offline mean of an arm is close to its online counterpart, the offline data should be more trusted. Conversely, if the discrepancy between the two is large, the algorithm should rely primarily on online feedback.

Based on this intuition, we construct $\mathrm{UCB}_t^{\mathrm{S}}(i)$ using a weighted empirical mean and a bias-adjusted confidence radius (Line 7 and 9). The empirical mean aggregates offline and online samples proportionally to their counts, while the confidence radius consists of two components: a standard deviation term based on the total offline and online sample size $N_i + T_i$, and a bias penalty scaled by the discrepancy bound $V_i$. The weight $N_i/(N_i + T_i)$ ensures that the penalty becomes more prominent as more offline data is used.

---

**Algorithm 1** Hybrid CUCB with Computation Oracle

---

**Require:** Valid bias bound $V$, number of arms $m$, offline data $\mathcal{B} := \{N_i, \{Y_{i,s}\}_{s=1}^{N_i}\}_{i=1}^m$, horizon $T$, Oracle

1: **for** each arm $i \in [m]$ **do**
2:     $\hat{\mu}_i^{\text{off}} \leftarrow \frac{1}{N_i} \sum_{s=1}^{N_i} Y_{i,s}, \quad T_i \leftarrow 0, \quad \hat{\mu}_i^{\text{on}} \leftarrow 0$
3: **end for**
4: **for** $t = 1, 2, \ldots, T$ **do**
5:     **for** each arm $i \in [m]$ **do**
6:         $\text{rad}_t(i) \leftarrow \sqrt{\frac{2\log(4mt^3)}{T_i}}$                                       $\triangleright = \infty$ if $T_i = 0$
7:         $\text{rad}_t^{\text{S}}(i) \leftarrow \sqrt{\frac{2\log(4mt^3)}{N_i+T_i}} + \frac{N_i}{N_i+T_i}V_i$                 $\triangleright = \infty$ if $N_i + T_i = 0$
8:         $\text{UCB}_t(i) \leftarrow \hat{\mu}_i^{\text{on}} + \text{rad}_t(i)$
9:         $\text{UCB}_t^{\text{S}}(i) \leftarrow \frac{N_i\hat{\mu}_i^{\text{off}}+T_i\hat{\mu}_i^{\text{on}}}{N_i+T_i} + \text{rad}_t^{\text{S}}(i)$
10:       $\bar{\mu}_i \leftarrow \min\left\{\text{UCB}_t(i),\, \text{UCB}_t^{\text{S}}(i),\, 1\right\}$
11:     **end for**
12:     $S \leftarrow \text{Oracle}(\bar{\mu}_1, \ldots, \bar{\mu}_m)$
13:     Play action $S$, triggering a set $\tau \subseteq [m]$ of base arms
14:     **for** each $i \in \tau$ with feedback $X_i^{(t)}$ **do**
15:         $T_i \leftarrow T_i + 1 \quad \hat{\mu}_i^{\text{on}} \leftarrow \hat{\mu}_i^{\text{on}} + (X_i^{(t)} - \hat{\mu}_i^{\text{on}})/T_i$
16:     **end for**
17: **end for**

---

Finally, by taking the minimum between the two UCB estimates, the algorithm can exploit useful offline data. Intuitively, if $N_i$ is large and $V_i$ is small such that $\text{UCB}_t^S(i) < \text{UCB}_t(i)$, then the offline data is useful for online exploration and the algorithm utilizes the hybrid $\text{UCB}_t^S(i)$. Otherwise, if $\mu_i^{\text{off}}$ and $\mu_i^{\text{on}}$ are far apart, then $\text{UCB}_t^S(i)$ becomes large. The algorithm would default to $\text{UCB}_t(i)$, effectively ignoring offline data. In both cases, the selection rule ensures that the decision is made conservatively, based on the estimated trustworthiness of the offline data. We next provide the regret upper bound for Algorithm 1 in Section 5.

## 5 THEORETICAL ANALYSIS

In this section, we provide the theoretical results for hybrid CUCB. We first provide the gap-dependent regret upper bound and the corresponding discussions. The gap-independent regret analysis comes later. The complete proof is provided in the appendix.

### 5.1 GAP-DEPENDENT BOUND

We first define the reward gaps used in the regret analysis.

**Definition 2** (Gap (Wang & Chen, 2017))**.** *Fix a distribution $D$ and its expectation vector $\boldsymbol{\mu}$. For each action $S$, we define the gap $\Delta_S = \max(0, \alpha \cdot opt_{\boldsymbol{\mu}} - r_S(\boldsymbol{\mu}))$. For each arm $i$, we define*

$$\Delta_{\min}^i = \inf_{S \in \mathcal{S}: p_i^{D,S}>0, \Delta_S>0} \Delta_S, \quad \Delta_{\max}^i = \sup_{S \in \mathcal{S}: p_i^{D,S}>0, \Delta_S>0} \Delta_S.$$

*As a convention, if there is no action $S$ such that $p_i^{D,S} > 0$ and $\Delta_S > 0$, we define $\Delta_{\min}^i = +\infty$, $\Delta_{\max}^i = 0$. Further define $\Delta_{\min} = \min_{i \in [m]} \Delta_{\min}^i$, $\Delta_{\max} = \max_{i \in [m]} \Delta_{\max}^i$.*

Let $\tilde{S} = \{i \in [m] \mid p_i^{\boldsymbol{\mu},S} > 0\}$ be the set of arms that could be triggered by $S$. Let $K = \max_{S \in \mathcal{S}} |\tilde{S}|$. To formally capture the influence of the discrepancy between offline and online environment, we introduce a measure $\omega_i := V_i + \mu_i^{\text{off}} - \mu_i^{\text{on}}$, $i \in [m]$. By the definition of $V$, we have that $\omega_i \in [0, 2V_i]$. Intuitively, the quantity $\omega_i$ allows us to express how much the offline data for arm $i$ deviates from the true online behavior, and plays a key role in determining the extent to which the offline data influences the online learning.

**Theorem 1** (Gap-Dependent Regret Bound). *For an H-CMAB-T problem $([m], \mathcal{S}, \mathcal{D}, D^{trig}, R, \mathcal{B})$ that satisfies monotonicity (Condition 1) and TPM bounded smoothness (Condition 2), the hybrid CUCB algorithm with an input bias control vector $V$ and an $(\alpha, \beta)$-approximation oracle achieves an $(\alpha, \beta)$-approximate gap-dependent regret bounded by:*

$$\text{Reg}_{\mu^{on}, \alpha, \beta}(T) \leq \sum_{i \in [m]} \max\left\{\frac{64\sqrt{2}B^2 K \log(4mT^3)}{\Delta^i_{\min}} - 8B\sqrt{2N'_i \log(4mT^3)}, 0\right\} + 4Bm + \frac{\pi^2}{6}\Delta_{\max},$$

(1)

*where*

$$N'_i = N_i \cdot \max\left\{1 - \frac{2BK\omega_i}{\Delta^i_{\min}}, 0\right\}^2.$$

Following Theorem 1, we now provide a detailed interpretation of the regret bound and its implications for how offline data is used by our algorithm.

A key quantity in the bound is $N'_i$, which represents the amount of *effectively utilized* offline data for arm $i$. The multiplicative factor can be interpreted as the *utilization rate* of the offline data. For a fixed online learning setting, the term $2BK/\Delta^i_{\min}$ is constant, so the utilization rate increases as the discrepancy $\omega_i$ decreases. When the offline data is unbiased (i.e., $V_i = \omega_i = 0$), we have full utilization: $N'_i = N_i$. In contrast, when $\omega_i \geq \Delta^i_{\min}/(2BK)$, the utilization rate drops to zero, and the offline data is effectively ignored. This reflects our design intuition: offline data that closely matches the online environment should be trusted more and used more aggressively. The result of Theorem 1 recovers the result of CMAB-T (Wang & Chen, 2017) as a special case when $N'_i = 0$ for all $i$. The setting may correspond to the case where the offline data do not exist (i.e. $N_i = 0$ for all $i \in [m]$) or the case that the offline data is fully misaligned with the online environment.

In general, our regret bound takes the form of the traditional regret in a purely online setting plus a benefit term of order $O(-\sqrt{N'_i})$. One might wonder why the adjustment is of order $O(-\sqrt{N'_i})$ instead of $O(-N'_i)$ in Cheung & Lyu (2024), which subtracts a term proportional to the effective number of plays, roughly $N'_i$, times the per-play regret. This difference arises from the distinct analytical techniques used in the MAB and CMAB-T settings. In MAB, the regret can be directly decomposed by counting the number of times each sub-optimal arm is selected. Thus, the benefit from offline data is proportional to the number of these selections avoided. In contrast, the CMAB-T analysis—enabled by the monotonicity and TPM condition—bounds the regret by analyzing the discrepancy between the UCB estimates and the true mean rewards. Intuitively, $O(-\sqrt{N'_i})$ comes from the regret saved in this discrepancy. With the offline data, we can interpret the online learning process as beginning from the $(N'_i + 1)$-th observation for each arm $i$. The resulting saving in the discrepancy between the UCB estimates and the true mean rewards is approximately $\sum_{s=1}^{N'_i} \sqrt{\log(4mT^3)/s} = O(\sqrt{N'_i \log(4mT^3)})$. When $N'_i$ is larger than $64B^2 K^2 \log(4mT^3)/(\Delta^i_{\min})^2$, the regret incurred during the online phase becomes bounded by a constant independent of $T$. This aligns with the same observations in the reduced MAB setting discussed in Cheung & Lyu (2024).

## 5.2 GAP-INDEPENDENT BOUND

We then analyze the gap-independent regret upper bound. We obtain two candidate bounds, denoted as $\psi$ and $\gamma$, each derived from a different proof technique. The final regret bound takes the minimum among them.

**Theorem 2** (Gap-Independent Regret Bound). *For an H-CMAB-T problem $([m], \mathcal{S}, \mathcal{D}, D^{trig}, R, \mathcal{B})$ that satisfies monotonicity (Condition 1) and TPM bounded smoothness (Condition 2), the hybrid CUCB algorithm with an input bias control vector $V$ and an $(\alpha, \beta)$-approximation oracle achieves an $(\alpha, \beta)$-approximate gap-independent regret bounded by:*

$$\text{Reg}_{\mu^{on}, \alpha, \beta}(T) \leq \min\{\psi, \gamma\} + 4Bm + \frac{\pi^2}{6}\Delta_{\max},$$

(2)

*where $\psi$ and $\gamma$ are two candidate bounds derived via distinct proof techniques:*

$$\psi = 8\sqrt{2}B\sqrt{\log(4mT^3)}\left(\sum_{i \in [m]} \max\left\{\sqrt{\frac{KT}{m}} - \sqrt{N''_i}, 0\right\} + \sqrt{mKT}\right),$$

(3)

$$\gamma = 16BKT\sqrt{\frac{2\log(4mT^3)}{\tau_*}} + BKT\omega_{\max}. \tag{4}$$

*Here*

$$N_i'' = N_i \cdot \max\left\{1 - \frac{\omega_i}{4\sqrt{2}}\sqrt{\frac{KT}{m\log(4mT^3)}}, 0\right\}^2, \quad \omega_{\max} = \max_i \omega_i, \tag{5}$$

*and $\tau_*$ is defined via*

$$\max_{\tau,\, n} \quad \tau$$
$$s.t. \quad \tau \le N_i + n(i) \text{ where } \tau \in \mathbb{N}, n(i) \in \mathbb{N}, \forall i,$$
$$\sum_{i \in [m]} n(i) \le KT.$$

These two upper bounds capture different aspects of how offline data can reduce exploration cost in the H-CMAB-T setting. We will interpret each bound, compare their relative strengths, and highlight how they recover or generalize existing results in the literature as follows.

Formally, the first bound $\psi$ involves the quantity $N_i''$, defined analogously to $N_i'$ in the gap-dependent bound, and it is interpreted as the amount of *effectively used* offline data. Similarly, the quantity $N_i''$ embodies the guiding principle behind our algorithmic design in Section 5.1: the more aligned the offline data is with the online environment, the more confidently and extensively it can be incorporated into the learning process. The setting where $N_i'' = 0$ for all $i$ recovers the pure online CMAB-T problem in (Wang & Chen, 2017), and the resulting bound matches their gap-independent result in order. In this sense, $\psi$ generalizes their analysis by quantifying the potential reduction in regret due to informative offline data via an $O(-\sqrt{N_i''})$ saving term. Moreover, it is worth noting that the use of the $\max\{\cdot, 0\}$ operator implies that $\psi$ ranges between a best-case value (when $N_i''$ is so large that the $\max\{\cdot, 0\} = 0, \forall i$) and a worst-case value (when $N_i'' = 0, \forall i$) matching the pure online regret bound. Specifically, $\psi$ lies between $8B\sqrt{mKT\log(4mT^3)}$ and $16B\sqrt{mKT\log(4mT^3)}$, depending on the informativeness of the offline data. Therefore, although $\psi$ reflects meaningful offline benefits and can cut down half of the regret at the best case, it does not improve the regret order corresponding to the specific problem parameters.

The second bound, $\gamma$, is derived via a relaxation of exploration constraints into a covering linear program. The LP solution $\tau_*$ appearing in $\gamma$ satisfies a uniform lower bound $\tau_* \ge KT/m$, which ensures that the first term in $\gamma$ is always at most the worst case of $\psi$. It can still be smaller when $N_i$ is large and $w_{\max}$ is small. In some extreme cases where $w_{\max} \le 1/BKT$ and $N_i \ge (BKT)^2\log(4mT^3)$, the bound $\gamma$ tends to be of constant order which is independent of $T$, highlighting the potential for offline data to fully eliminate exploration cost under perfect alignment. Moreover, $\gamma$ structurally aligns with recent work on leveraging offline data in the classical MAB setting (Cheung & Lyu, 2024). By setting $K = B = 1$, our H-CMAB-T problem reduces to a hybrid MAB scenario. In this special case, $\gamma$ recovers (and slightly tightens) Cheung & Lyu (2024): their bound includes a saving term of the form $2TV_{\max}$, whereas ours uses $Tw_{\max}$ with $w_{\max} \le 2V_{\max}$.

We now compare the two bounds in terms of tightness and interpretability. The bound $\psi$ provides a uniform guarantee and reflects a conservative lower baseline. While it never diverges, it also does not yield a tighter rate even when offline data is abundant. In contrast, $\gamma$ can become substantially tighter in favorable regimes. When the offline data is highly informative (i.e., large $N_i$ and small $\omega_i$), $\gamma$ can reduce the regret significantly. For example, in the ideal case of $N_i \ge (BKT)^2\log(4mT^3)$ and $\omega_{\max} \le 1/BKT$, the bound tends to be a constant, matching our expectation that regret should vanish when offline information fully resolves arm uncertainty.

Together, these two bounds form a comprehensive characterization of the gap-independent regret in H-CMAB-T. They offer different trade-offs between robustness, interpretability, and tightness, and demonstrate how the size, bias, and coverage of offline data influence the learning performance

## 6 EXPERIMENTS

In this section, we compare our proposed hybrid CUCB with existing CUCB for the pure online setting (Wang & Chen, 2017) and CLCB for the pure offline setting (Liu et al., 2025). To evaluate the performance of CLCB, we first use this algorithm to select an action based on the offline data set and always select this action in the following rounds. For simplicity, we assume that $N_i = N$ and $V_i = V$ for any arm $i$. Due to the space limit, more details about the reward function and triggering mechanism, as well as the experimental setting and real-world validations, are deferred to appendix.

We evaluate on unbiased offline datasets with varying sizes $N \in \{10, 50, 200\}$. As shown in Figure 1, hybrid CUCB consistently outperforms both online CUCB and offline CLCB. The improvement stems from the warm-start provided by offline data, which reduces early exploration. The advantage becomes more pronounced with larger $N$, and when $N$ is sufficiently large (e.g., $N = 200$), hybrid CUCB achieves constant regret. Compared to CLCB, the hybrid approach is especially superior when offline data is scarce, since CLCB relies solely on potentially inaccurate offline estimates.

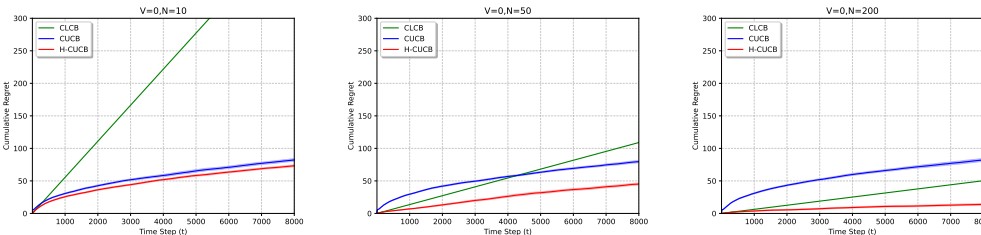

Figure 1: Performance comparison of hybrid CUCB against baselines with unbiased offline data set.

We further evaluate the robustness of the algorithms under distributional bias between the offline and online environments. Specifically, we consider varying levels of bias $V \in \{0.2, 0.3, 0.4\}$, assuming a sufficiently large offline dataset size ($N = 200$) to ensure reliable offline estimates. The results, presented in Figure 3, demonstrate that our hybrid CUCB algorithm consistently outperforms or matches the baseline performance across all tested levels of distributional bias.

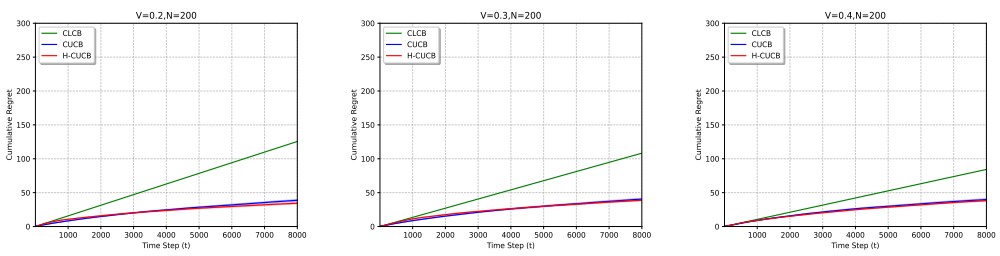

Figure 2: Performance comparison of hybrid CUCB against baselines with the biased offline data set.

## 7 CONCLUSION

We introduce H-CMAB-T, a new framework that extends classical CMAB-T by incorporating available offline data into online learning. We propose the hybrid CUCB algorithm, which selectively leverages offline observations via a minimum of two confidence bounds, controlled by a bias-aware mechanism. Theoretically, we established both gap-dependent and gap-independent regret bounds, showing that our method effectively reduces exploration through a data-dependent saving term. Empirical results further corroborate our theoretical findings, demonstrating the effectiveness of the proposed method in benchmark CMAB-T scenarios. The current CMAB-T framework does not naturally handle high-dimensional contexts or side information. Extending hybrid learning to contextual CMAB-T represents a promising direction, with potential for broader applicability in practical scenarios.

ETHICS STATEMENT

No ethical concerns.

REPRODUCIBILITY STATEMENT

We have provided detailed descriptions of our algorithms, theoretical analysis, and experimental settings in the main text and the appendix. All hyperparameters and implementation details (including offline data generation, action set construction, and evaluation protocol) are specified. Since our experiments are small-scale numerical simulations or test on a small subset of real world data set, all details are explicitly described in the text and appendix, making the results straightforward to reproduce without releasing code.

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
