# OpenReview forum: "Hybrid Combinatorial Multi-armed Bandits with Probabilistically Triggered Arms"
_ICLR.cc/2026/Conference — ICLR 2026 Conference Withdrawn Submission_

### Official Review · Reviewer_CBtw · 2025-10-25

**Soundness:** 2
**Presentation:** 3
**Contribution:** 2
**Rating:** 4
**Confidence:** 3

**Summary:**

The paper studies combinatorial multi-armed bandits with probabilistically triggered arms (CMAB-T) in a hybrid learning setup with access to offline data. The offline data can have biased arm means, making its utilization challenging for fast learning. They remedy this by proposing an algorithm, Hybrid (H)-CUCB, which, per arm, takes the minimum of (i) a standard online CUCB upper bound and (ii) a bias-aware hybrid UCB that mixes offline+online samples and adds a bias penalty. Under monotonicity and 1-norm TPM bounded-smoothness, the paper proves gap-dependent and gap-independent regret upper bounds with that depend on the effective, bias-adjusted offline sample sizes. Experiments on synthetic CMAB-T learning-to-rank and a small MovieLens split show H-CUCB improves over CUCB (online) and an offline baseline (CLCB).

**Strengths:**

1. To the best of my knowledge, this paper provides the first hybrid (offline+online) treatment for general CMAB-T with probabilistic triggering, giving regret bounds that reduce to online CUCB when offline data are unhelpful and improve with aligned offline data.

2. The decomposition of the regret clearly shows the benefit of using offline data via its effective number of samples, resulting in $O(-\sqrt{N’_i})$ savings in Theorem 1. Discussions of Theorems 1 and 2 explain how the upper bounds on regret behave under different regimes of offline data size and quality.

3. The significance resides in addressing multiple relevant applications (such as influence maximization or learning-to-rank) where offline data is common but usually biased.

**Weaknesses:**

1. While the authors make connections between their general results and the edge cases with fully informative offline data and non-informative offline data, they provide no new lower bounds on the regret. This negatively impacts the significance of the work since it is mainly theoretical in nature, and related works of this kind usually have accompanying lower bounds.

2. A limitation is that the algorithm requires the knowledge of an upper bound on $V$, the discrepancy of the means of online and offline data. In practice, it will be possible to estimate this discrepancy and adjust the algorithm as it runs. While this may make obtaining theoretical regret bounds challenging (the authors mention this in Remark 1), understanding the effect of online estimation of this quantity on the performance can be carried out via simulations.

3. Experimental results are not comprehensive. Standard deviations of the regrets are not plotted. Sensitivity to misspecifications on the bias bound is not investigated.

4. The analysis largely adapts the CMAB-T proof toolkit and tracks how offline bias enters the confidence radii. From this point of view, the technical novelty is incremental.

**Questions:**

1. Uniform lower bound on $\tau^*$ in line 414: Please elaborate on its correctness and point to the proof.

2. Experiments: The figure do not demonstrate constant regret. Can you elaborate on the constant regret result?

3. Can the benefit order in Theorem 1 be improved to the one in Cheung & Lyu (2024), perhaps using a different bounding technique? Shouldn't the regret terms match (at least) when the CMAB-T formulation is reduced to a standard bandit?

---

> ### Author Response · Authors · 2025-11-26
> **Rebuttal part1**
>
> We thank the reviewer CBtw for providing valuable feedback. Please see our response below.
>
> ----
>
> **Lower Bound**
>
> In the revised PDF, we have added a full and rigorous proof in Appendix E.
> Here we summarize the key idea and construction.
>
> We consider a CMAB-T gating instance with triggering probabilities
> $p_{S,i} = 1/K$ and linear expected reward
> $r_S(\mu) = \frac{B}{K} \sum_{i \in S} \mu_i$.
>
> For each suboptimal arm $i$, we construct a neighboring environment $\nu^{(i)}$
> in which only $\mu_i$ is increased by $\Delta$, while keeping the offline
> distributions as close as possible so that the algorithm cannot easily
> distinguish the two.
>
> By a standard change-of-measure argument, the online and offline observations
> must satisfy
> $$
> \mathbb{E} _ {\mu}[T_i]\Theta(\Delta^2)+N_i(\Delta - 2V_i) _ {+}^{2} \ge\log T .
> $$
>
> Solving for $\mathbb{E} _ {\mu}[T_i]$ and substituting it into the regret identity
> $\mathrm{Reg} _ i(T)= B\Delta_i\mathbb{E}_{\nu}[T_i]$ gives
> $$
> \mathrm{Reg}_i(T) \ge B\left(\frac{\log T}{\Delta_i}-\sqrt{N_i''\log T}\right),
> \qquad
> \text{with $\Delta_i$ chosen optimally.}
> $$
>
> Summing over all suboptimal arms yields a lower bound with the same structure
> as our upper bound.  All intermediate derivations are provided in the revised
> Appendix E.
>
> ----
>
> **Unknown bias or bias bound $V$ ?**
>
> We agree that relaxing the dependence on the hyperparameter $V$ would be more desirable. We do not have an ideal method yet but glad to have a discussion on it.
>
> **1.estimating the bias** A natural idea is to attempt to learn $V$ during the online learning process. However, this introduces a fundamental trade-off between the cost of learning the bias bound $V$ and the potential benefit of incorporating offline data.
>     Specifically, when $V$ is small, accurately estimating the bias bound requires a large number of online samples—potentially far exceeding the exploration cost needed to identify and eliminate sub-optimal arms. In this case, the overhead of estimating $V$ may outweigh the benefit of leveraging offline data. On the other hand, if $V$ is large, the discrepancy between offline and online estimates can be detected quickly, requiring relatively few online interactions. But in such cases, the offline data is highly biased and therefore not very useful for improving performance, so we would naturally fall back on a pure online strategy.
>     From a statistical perspective, controlling the variance of the estimator $V_i =\mu_i^{\mathrm{off}} - \mu_i^{\mathrm{on}}$
>     necessarily requires controlling the estimation errors of $\mu_i^{\mathrm{off}}$ and $\mu_i^{\mathrm{on}}$ *simultaneously*.  Neither component can be omitted.  This implies that such an approach would require both:
>
> (a)stronger assumptions (or size requirements) on the offline data in order to control the error of the offline estimator, and
>
> (b)sufficiently accurate online learning of $\mu_i^{\mathrm{on}}$.
>
> Consider an extreme scenario where the offline dataset is so large that $\mu_i^{\mathrm{off}}$ is essentially known. Even in this case, meaningful control of $V_i$ would still depend on accurately learning $\mu_i^{\mathrm{on}}$ from online interactions. If estimating $V_i$ well requires us to learn $\mu_i^{\mathrm{on}}$ almost as accurately as in the fully online setting, then focusing on $V_i$ as a primary target becomes less attractive for regret minimization.
>
> As a result, instead of aiming solely to minimize online regret, one could explicitly incorporate the accurate estimation of the bias vector $V$ into the objective. A multi-objective formulation that trades off regret and the estimation error of $V$ may offer a richer perspective.
>
> **2.Parallel online vs. hybrid routes.**
>     Alternatively, one might avoid directly estimating $V$, and instead run two learning routes in parallel:
> (i) a purely online algorithm that guarantees a stable baseline performance, and
> (ii) a hybrid algorithm that leverages the offline data.
>
> Based on the feedback of the two routes, the learner can adaptively decide whether to place more weight on the pure online path or on the hybrid path as learning proceeds. As discussed above on the problems we would meet to learn the true bias $V$, this is another important future work for us.

---

> ### Author Response · Authors · 2025-11-26
> **Rebuttal part2**
>
> ----
>
> **Experiments**
>
> (1)Indeed, warm-starting CUCB using offline CLCB estimates is a natural hybrid baseline and has been used in prior work without distributional shift.
> Our focus in this paper is on the more challenging setting with *biased* offline data, where directly warm-starting CUCB using possibly-shifted offline estimates can introduce additional bias.
> Nevertheless, we agree that including this baseline would strengthen the empirical section.
> We will incorporate a warm-start CUCB baseline (which in fact is the H-CUCB but always setting the input $V=0$) in the future version.
> Our algorithmic design is fully compatible with this baseline, and we expect its performance to be worse than CUCB and our proposed hybrid method.
>
> (2)In the current submission, the standard deviation bands were included but were relatively small and therefore visually difficult to distinguish in the plotted scale.
> We agree that making the variability more clearly visible would improve the readability of the figures in future version.
>
> ----
>
> **Novelty**
>
> While deriving UCB-based algorithms is common in the bandit literature, adapting such methods to the CMAB-T setting with hybrid feedback introduces new challenges. Below, we highlight the key differences in both analytical techniques and theoretical results in comparison to [1].
>
> **Analytical techniques.**
> The main challenge in the hybrid CMAB-T setting lies in deciding, at each round, which UCB estimate to use: the one based on biased offline data or the one from online interactions. In the simpler MAB setting studied by [1], this decision is easier because the regret can be directly decomposed using suboptimality gaps $\Delta_i$ and the number of arm pulls. This decomposition enables a relatively straightforward threshold analysis between hybrid UCB estimates and the optimal mean $\mu^*$.
> However, in CMAB-T, such a gap-based regret decomposition does not apply. Instead, regret must be controlled through confidence intervals and triggering probabilities, which do not naturally lend themselves to closed-form thresholds. A naïve approach would be to derive a switching rule by solving the inequality between the two UCB estimates. Unfortunately, this leads to high-degree, analytically intractable expressions that lack interpretability.
>
> To address this challenge, we propose a principled decision rule that is tightly coupled with the uncertainty-based structure of CMAB-T analysis. The core idea is to incorporate offline data only when the reduction in uncertainty it brings outweighs the potential error due to bias. Specifically, we analyze arm-level bias and aggregate it at the super-arm level via the triggering mechanism, and discuss the values of such aggregated bias and the per-round regret. This approach yields a regret analysis that naturally balances robustness and adaptivity, and it reveals how offline data can be selectively leveraged to improve performance without compromising the theoretical guarantees of pure online learning.
>
> **Theoretical results.** Our theoretical results also reflect a fundamental difference from those in [1]. While [1] establishes a regret bound of the form $O({m \log T}/{\Delta} - \sum_i N'_i \Delta_i)$, our bound takes the form $O({m \log T}/{\Delta} - \sum_i \sqrt{N_i' \log T})$. The saving term in [1] intuitively captures the reduced number of times suboptimal arms are selected. In contrast, our saving term reflects the reduction in uncertainty enabled by the presence of offline data—highlighting a distinct interpretation and analytical perspective.
>
> Uncertainty-based analysis plays a central role in many complex learning problems, such as linear bandits and linear MDPs. In contrast, per-arm analysis lacks this adaptability and does not extend naturally to such settings. We believe that our analytical approach, grounded in the trade-off between uncertainty and bias, can be generalized to a broader class of learning problems.
>
> ----
>
> **Why $\tau^{*} \ge KT/m$**
>
> Intuitively, even in the most conservative case where we ignore all initial counts
> and take $N_i = 0$ for every arm $i$, we still have at most $KT$ total pulls to
> distribute across $m$ arms.  Distributing these pulls as evenly as possible implies
> that at least one feasible allocation satisfies
> $$
> \tau \le N_i + n(i) = n(i) \approx \frac{KT}{m} \quad \text{for all } i,
> $$
> so the optimal value $\tau_\ast$ cannot be smaller than (up to integer rounding)
> $KT/m$.  In particular, this yields a uniform lower bound
> $\tau_\ast \gtrsim KT/m$ independent of the specific values of $N_i$.
>
> A more rigorous derivation of this bound, including the precise treatment of
> integer effects (using $\lfloor KT/m \rfloor$), can be provided in the appendix
> if needed.
>
> ----

---

> ### Author Response · Authors · 2025-11-26
> **Rebuttal part3**
>
> **Reduces to a standard bandit**
>
> As discussed in our novelty section, our analysis is carried out in the more general
> CMAB-T setting, which is harder than the standard multi-armed bandit (MAB): it
> involves combinatorial actions and probabilistically triggered arms.
> Techniques developed for CMAB-T can in principle be specialized to the MAB case
> (and hence could be used to recover hybrid-MAB type results similar in spirit to ours),
> but the reverse direction does not generally hold.
>
> Regarding the benefit order, we note that the sample complexity requirement on the
> effective offline sample size $N_i'$ for achieving *constant* regret is of the
> same order as in [1]: in both works one needs $N_i' = \Theta(T^2)$
> (up to logarithmic factors).  In this sense, the scaling with respect to the offline
> dataset size is consistent between our hybrid CMAB-T analysis and the hybrid MAB setting.
>
>
> *[1]Wang Chi Cheung and Lixing Lyu. Leveraging (biased) information: Multi-armed bandits with
> offline data. arXiv preprint arXiv:2405.02594, 2024. doi: 10.48550/arXiv.2405.02594. Accepted
> to ICML 2024.*

---

> > ### Comment · Reviewer_CBtw · 2025-11-27
> >
> > Thanks for the clarifications regarding the lower bound and upper bounds. The new lower bound analysis improves the contribution. I will update my score accordingly.

---

> > > ### Author Response · Authors · 2025-11-27
> > >
> > > Thank you very much for the constructive feedback and for updating the score. We truly appreciate your time and effort.
> > > Please feel free to let us know if there are any additional questions or
> > > clarifications needed — any comments or further questions are valuable for improving our work.

---

### Official Review · Reviewer_AxUf · 2025-11-01

**Soundness:** 3
**Presentation:** 2
**Contribution:** 2
**Rating:** 4
**Confidence:** 3

**Summary:**

This paper addresses the combinatorial multi-armed bandit problem with probabilistically triggered arms (CMAB-T), focusing on integrating the benefits of offline data and online interaction (exploration, adaptability). The authors formulate the hybrid CMAB-T (H-CMAB-T) setting, introduce a hybrid CUCB algorithm that leverages both data sources by adaptively trading off their contributions with a bias-aware dual-UCB mechanism, and provide theoretical regret guarantees. Empirical evaluations on synthetic and real-world datasets demonstrate gains over purely online and purely offline baselines, especially in regimes with limited or biased offline data.

**Strengths:**

- The hybrid learning problem in CMAB-T is relevant and underexplored. This paper provides a systematic formalization, bridging existing gaps in purely online and purely offline settings.
- Both gap-dependent (Theorem 1) and gap-independent (Theorem 2), regret bounds are derived.
- Results in Figures 1 and 2 show consistent improvement of hybrid CUCB over CUCB (online) and CLCB (offline) across different offline data sizes and bias settings.

**Weaknesses:**

1. Mostly small-scale synthetic + one real dataset; no stress tests on larger action spaces/trigger structures or runtime.
2. The paper hypes the hybrid approach’s advantage but gives limited attention to scenarios where hybridization fails (i.e., more adverse bias structures, very small offline samples, or worst-case triggering).

**Questions:**

1. How does the proposed hybrid CUCB perform when the offline data shows significant non-uniform coverage or is adversarially biased for specific arms/actions? Is there a regime where offline data systematically degrades performance or slows learning?
2. What is the computational overhead of hybrid CUCB relative to standard CUCB, especially in large-scale or high-dimensional action spaces?
3. Are there settings where hybrid CUCB underperforms compared to pure online learning due to pathological offline data or wrong $V$ settings? The current discussion is mostly positive or neutral.

---

> ### Author Response · Authors · 2025-11-26
> **Rebuttal to reviewer AxUf**
>
> We thank the reviewer AxUf for providing valuable feedback. Please see our response below.
>
> ----
>
> **Experiments**
>
> The present paper focuses primarily on the theoretical aspects of hybrid learning under distributional shift, and for this reason our experimental design was intentionally kept minimal to isolate the key phenomena predicted by the theory.
>
> That said, we fully agree that evaluating the method on larger action spaces and more complex triggering structures would strengthen the empirical section.
> In the future version of the paper, we plan to supplement our results with:
>
> （1）experiments on **Online Influence Maximization (OIM)**, a standard large-scale CMAB-T problem with richer triggering structures, and
>
> （2）additional experiments on **larger real-world datasets** to further validate the robustness of our method.
>
> ----
>
> **Scenarios where hybrid methods fail**
>
> Although the main text did not emphasize failure modes, we note that Theorem 1 already reveals how hybridization can degrade under adverse conditions.
>
> **(i) Limited offline data.**  In the regret bound, the contribution of the offline dataset appears through terms of the form $-\sqrt{N_i} ~ \text{or} -\sqrt{N_i'},$
> which explicitly shows that small offline sample sizes yield little benefit.
> When $N_i$ is small, the negative contribution is weak, and the hybrid method behaves close to the purely online algorithm, as expected.
>
> **(ii) Adverse bias structures.**  The effective sample size  $N_i' = N_i  \min ( 1 - \frac{2 B K w_i}{\Delta_i},0 )^{2}$ makes the dependence on the bias structure clear. Since  $w_i = V_i + \mu_i^{\mathrm{off}} - \mu_i^{\mathrm{on}},$
> larger $w_i$ decreases $N_i'$, meaning that the offline data provide *less* useful guidance to the online learner.
> In extreme cases, a large $w_i$ can remove almost all benefit from the offline data, causing the hybrid method to revert to online performance.
>
> **(iii) When offline data make bad arms look even worse, or when with bias but wrong input $V_i$.**
> Interestingly, when $\mu_i^{\mathrm{off}} < \mu_i^{\mathrm{on}}$, even with a bias term $V_i$, it is possible that $w_i$ becomes small. In this scenario, the offline data actually reinforce that certain arms are suboptimal, which $accelerates$ online exploration and improves regret.
> This asymmetry---hybrid help vs. hybrid harm---is fully reflected in the structure of $w_i$. On the other hand, however, if $V_i$ is close to zero but $\mu_i^{\mathrm{off}} > \mu_i^{\mathrm{on}}$, the algorithm will think the offline data is informative and lead to worse performance than pure online learning.
> The guarantee that our hybrid method never performs worse than the pure online baseline relies on the assumption that the provided values $V_i$ are valid upper bounds on the true biases. If no prior information about the bias is available, one may simply set $V_i = 1$, since $V_i$ is used as a conservative upper bound rather than an accurate estimate. Under this choice, the robustness guarantee of our method remains intact.
>
> Taken together, Theorem 1 already characterizes both the beneficial and detrimental regimes of hybridization. Moreover, since we also setting the pure online UCB in algorithm,
> \(when bias is large the algorithm will ignore the hybrid UCB\), its performance would never worse than pure online, which we can also check this in Th 1.
>
> ----
>
> **Computational overhead of hybrid CUCB relative to standard CUCB**
>
> In our setting, the computational complexity of CMAB-T algorithms is dominated by
> the structure of the triggering process and the oracle used to select a super-arm.
> These components are identical for CUCB and H-CUCB.
>
> The hybrid variant introduces only a minimal additional cost: in each round,
> for each arm $i$, we compute an additional UCB term based on the offline estimator.
> This adds only $O(m)$ extra arithmetic operations per round, where $m$ is the
> number of base arms. No additional combinatorial optimization or triggering
> simulation is required beyond what CUCB already performs.
>
> Therefore, the computational overhead of H-CUCB compared with standard CUCB is
> negligible and scales linearly with the number of arms. In large-scale or
> high-dimensional action spaces, the dominant runtime remains the same oracle
> and triggering operations shared by both methods.

---

> > ### Comment · Reviewer_AxUf · 2025-11-27
> >
> > I thank the authors for their detailed response. They addressed my computational overhead and robustness concerns. While  I note the authors' promise to include OIM and larger datasets in the final version, as these were not provided in the rebuttal, the empirical evaluation remains thin and limited to small-scale/synthetic settings.

---

> > > ### Author Response · Authors · 2025-11-28
> > >
> > > Thank you very much for the follow-up and for acknowledging our responses.
> > >
> > > We would like to clarify that the revised PDF uploaded focuses only on the newly added lower bound and its full proof, as suggested in the reviews. The additional OIM experiments and larger-scale evaluations are currently in progress.
> > >
> > > If we are not able to finish these experiments before the rebuttal deadline,
> > > we will explicitly state their status in the final comments.
> > >
> > > We sincerely appreciate your constructive feedback and the opportunity to
> > > further improve the paper.

---

### Official Review · Reviewer_VxCG · 2025-11-02

**Soundness:** 2
**Presentation:** 2
**Contribution:** 2
**Rating:** 4
**Confidence:** 3

**Summary:**

The paper introduces Hybrid CMAB-T (H-CMAB-T), extending classical combinatorial multi-armed bandits with probabilistically triggered arms (CMAB-T) to a hybrid learning regime. It proposes Hybrid-CUCB, which forms two arm-level UCBs, including a pure online CUCB estimate and a bias-aware hybrid estimate that pools  samples while penalizing possible offline/online mismatches via a known per-arm bias bound $V_i$. The algorithm feeds the coordinate-wise minimum of these bounds to an $(\alpha,\beta)$-approximation oracle over the action space. Theoretically, the paper proves both gap-dependent and gap-independent regret bounds that interpolate between pure online CMAB-T and (warm-started) hybrid learning; the savings scale with an effective amount of usable offline data that depends on the size and bias of the log. Empirically (synthetic and small-scale), Hybrid-CUCB improves over online CUCB and a one-shot offline policy, and remains robust under moderate bias.

**Strengths:**

1. The setting for hybrid CMAB-T is naturally and clean.

2. By following prior papers and combining algorithms together, the Hybrid CUCB works naturally and well, validated both theoretically and empirically.

3. The interpretations and intuitions of saving samples for terms $N_i^{\prime}$ and $N_i^{\prime \prime}$ are clear and good to be understood.

4. Provide both gap-dependent bound and gap-independent bound, comprehensive in the upper bound results.

**Weaknesses:**

1. Lacking of lower bound: This paper provides promising upper bounds including gap dependent version and independent version. However, no lower bound is provided. In [1], lower bounds for both instance-dependent and instance-independent versions are given, and the optimality is guaranteed. I treat this part of contribution as crucial in their work since it is crucial to see how effective of these offline samples can really function as, at most, for the biased hybrid setting. By only providing the theoretical upper bounds, we can only understand how effective these offline samples are for the proposed algorithm, but not helpful for us to understand the real insight behind the problem setting (e.g., whether the hybrid CMAB-T problem theoretical results have exactly similar versions of hybrid MAB results, as [1] provides). The authors give some intuitions for it (such as from line 348-line 361), which is good. But I believe further detailed and promising interpretations and formal guarantees are really crucial for me.



2. Strong assumptions on known $V_i$ : The proposed algorithm Hybrid CUCB needs the knowledge of $V_i$ for calculating the biased estimate $\text{UCB}^{S}$, which is also the way did in [1]. However, I consider this is a really strong assumption in the works. Specifically, it is now realistic enough to know a very concrete bounded gap of bias between online and offline data, leading this to be a very strong assumption (most scenarios the gap can only be chosen to be infinity, meaning the offline data is totally useless for the online learning phase). I understand that in [1] they give an impossibility result on cases when there does not exist a bounded bias guarantee for offline and online data. Although this is not given in this paper, I believe it is trivial to prove that. However, what I am curious about is: if there exists some gap (i.e. $V_i$ ) between online and offline data, but the gap is unknown, is there any theoretical results to estimate this gap and utilize the estimated ones rather than given ones to have some further results without dependency on the known $V_i$ ? I think this should be a crucial (but currently lacking) contribution in this line of works.

3. Problem novelty: Generally I treat this paper as considering the CMAB-T problem instead of MAB problem with the biased hybrid setting as [1]. To me, nothing is much special here and everything corresponds to the theoretical intuitions currently. I wonder if there is any unique challenge in CMAB-T+[1] compared to MAB+[1], and if so, the authors should clarify it more clearly. Further interpretations on technical novelty or challenges are of great importance in this paper.


[1] Leveraging (Biased) Information: Multi-armed Bandits with Offline Data

**Questions:**

For (1) and (2) in weaknesses, Can the authors provide some further formal results? If formally proving them are technically difficult or not ready, at least some comprehensive intuitions on the lower bound and cases without knowing Vi should be given to help readers understand the whole contribution, and what is lacking now. For (3) in weaknesses, can the authors better interpret the novelty of this problem and contributions, rather than a simple CMAB-T+[1]? I would increase my score if problems (1) and (2) are answered perfectly, while (3) can be interpreted more clearly and comprehensive.

---

> ### Author Response · Authors · 2025-11-26
> **Rebuttal part1**
>
> We thank the reviewer VxCG for providing valuable feedback. Please see our response below.
>
> ----
>
> **Lower Bound**
>
> In the revised PDF, we have added a full and rigorous proof in Appendix E.
> Here we summarize the key idea and construction.
>
> We consider a CMAB-T gating instance with triggering probabilities
> $p_{S,i} = 1/K$ and linear expected reward
> $r_S(\mu) = \frac{B}{K} \sum_{i \in S} \mu_i$.
>
> For each suboptimal arm $i$, we construct a neighboring environment $\nu^{(i)}$
> in which only $\mu_i$ is increased by $\Delta$, while keeping the offline
> distributions as close as possible so that the algorithm cannot easily
> distinguish the two.
>
> By a standard change-of-measure argument, the online and offline observations
> must satisfy
> $$
> \mathbb{E} _ {\mu}[T_i]\Theta(\Delta^2)+N_i(\Delta - 2V_i) _ {+}^{2} \ge\log T .
> $$
>
> Solving for $\mathbb{E} _ {\mu}[T_i]$ and substituting it into the regret identity
> $\mathrm{Reg} _ i(T)= B\Delta_i\mathbb{E}_{\nu}[T_i]$ gives
> $$
> \mathrm{Reg}_i(T) \ge B\left(\frac{\log T}{\Delta_i}-\sqrt{N_i''\log T}\right),
> \qquad
> \text{with $\Delta_i$ chosen optimally.}
> $$
>
> Summing over all suboptimal arms yields a lower bound with the same structure
> as our upper bound.  All intermediate derivations are provided in the revised
> Appendix E.
>
> ----
>
> **Unknown bias or bias bound $V$ ?**
>
> We agree that relaxing the dependence on the hyperparameter $V$ would be more desirable. We do not have an ideal method yet but glad to have a discussion on it.
>
> **1.estimating the bias** A natural idea is to attempt to learn $V$ during the online learning process. However, this introduces a fundamental trade-off between the cost of learning the bias bound $V$ and the potential benefit of incorporating offline data.
>     Specifically, when $V$ is small, accurately estimating the bias bound requires a large number of online samples—potentially far exceeding the exploration cost needed to identify and eliminate sub-optimal arms. In this case, the overhead of estimating $V$ may outweigh the benefit of leveraging offline data. On the other hand, if $V$ is large, the discrepancy between offline and online estimates can be detected quickly, requiring relatively few online interactions. But in such cases, the offline data is highly biased and therefore not very useful for improving performance, so we would naturally fall back on a pure online strategy.
>     From a statistical perspective, controlling the variance of the estimator $V_i =\mu_i^{\mathrm{off}} - \mu_i^{\mathrm{on}}$
>     necessarily requires controlling the estimation errors of $\mu_i^{\mathrm{off}}$ and $\mu_i^{\mathrm{on}}$ *simultaneously*.  Neither component can be omitted.  This implies that such an approach would require both:
>
> (a)stronger assumptions (or size requirements) on the offline data in order to control the error of the offline estimator, and
>
> (b)sufficiently accurate online learning of $\mu_i^{\mathrm{on}}$.
>
> Consider an extreme scenario where the offline dataset is so large that $\mu_i^{\mathrm{off}}$ is essentially known. Even in this case, meaningful control of $V_i$ would still depend on accurately learning $\mu_i^{\mathrm{on}}$ from online interactions. If estimating $V_i$ well requires us to learn $\mu_i^{\mathrm{on}}$ almost as accurately as in the fully online setting, then focusing on $V_i$ as a primary target becomes less attractive for regret minimization.
>
> As a result, instead of aiming solely to minimize online regret, one could explicitly incorporate the accurate estimation of the bias vector $V$ into the objective. A multi-objective formulation that trades off regret and the estimation error of $V$ may offer a richer perspective.
>
> **2.Parallel online vs. hybrid routes.**
>     Alternatively, one might avoid directly estimating $V$, and instead run two learning routes in parallel:
> (i) a purely online algorithm that guarantees a stable baseline performance, and
> (ii) a hybrid algorithm that leverages the offline data.
>
> Based on the feedback of the two routes, the learner can adaptively decide whether to place more weight on the pure online path or on the hybrid path as learning proceeds. As discussed above on the problems we would meet to learn the true bias $V$, this is another important future work for us.

---

> ### Author Response · Authors · 2025-11-26
> **Rebuttal part2**
>
> ----
>
> **Novelty**
>
> While deriving UCB-based algorithms is common in the bandit literature, adapting such methods to the CMAB-T setting with hybrid feedback introduces new challenges. Below, we highlight the key differences in both analytical techniques and theoretical results in comparison to [1].
>
> **Analytical techniques.**
> The main challenge in the hybrid CMAB-T setting lies in deciding, at each round, which UCB estimate to use: the one based on biased offline data or the one from online interactions. In the simpler MAB setting studied by [1], this decision is easier because the regret can be directly decomposed using suboptimality gaps $\Delta_i$ and the number of arm pulls. This decomposition enables a relatively straightforward threshold analysis between hybrid UCB estimates and the optimal mean $\mu^*$.
> However, in CMAB-T, such a gap-based regret decomposition does not apply. Instead, regret must be controlled through confidence intervals and triggering probabilities, which do not naturally lend themselves to closed-form thresholds. A naïve approach would be to derive a switching rule by solving the inequality between the two UCB estimates. Unfortunately, this leads to high-degree, analytically intractable expressions that lack interpretability.
>
> To address this challenge, we propose a principled decision rule that is tightly coupled with the uncertainty-based structure of CMAB-T analysis. The core idea is to incorporate offline data only when the reduction in uncertainty it brings outweighs the potential error due to bias. Specifically, we analyze arm-level bias and aggregate it at the super-arm level via the triggering mechanism, and discuss the values of such aggregated bias and the per-round regret. This approach yields a regret analysis that naturally balances robustness and adaptivity, and it reveals how offline data can be selectively leveraged to improve performance without compromising the theoretical guarantees of pure online learning.
>
> **Theoretical results.** Our theoretical results also reflect a fundamental difference from those in [1]. While [1] establishes a regret bound of the form $O({m \log T}/{\Delta} - \sum_i N'_i \Delta_i)$, our bound takes the form $O({m \log T}/{\Delta} - \sum_i \sqrt{N_i' \log T})$. The saving term in [1] intuitively captures the reduced number of times suboptimal arms are selected. In contrast, our saving term reflects the reduction in uncertainty enabled by the presence of offline data—highlighting a distinct interpretation and analytical perspective.
>
> Uncertainty-based analysis plays a central role in many complex learning problems, such as linear bandits and linear MDPs. In contrast, per-arm analysis lacks this adaptability and does not extend naturally to such settings. We believe that our analytical approach, grounded in the trade-off between uncertainty and bias, can be generalized to a broader class of learning problems.
>
> ----
>
> *[1]Wang Chi Cheung and Lixing Lyu. Leveraging (biased) information: Multi-armed bandits with
> offline data. arXiv preprint arXiv:2405.02594, 2024. doi: 10.48550/arXiv.2405.02594. Accepted
> to ICML 2024.*

---

### Official Review · Reviewer_9DtE · 2025-11-11

**Soundness:** 2
**Presentation:** 1
**Contribution:** 2
**Rating:** 2
**Confidence:** 3

**Summary:**

The authors consider a combinatorial multi-armed bandit problem with probabilistically triggered arms (CMAB-T) for which offline data is considered, but there may be a distributional shift wrt the online setting.  The authors propose a hybrid combinatorial CUCB method using bounds on the base arm biases and derive gap-dependent and gap independent bounds, in some special cases recovering results for simpler cases.

**Strengths:**

- The problem appears to be novel.  CMAB-T has been studied in purely online (multiple works) and recently in purely online.  There is also work for several bandit settings using hybrid data, though before Cheung and Lyu (2024) most/all considered identically distributed environments.

- The authors analyze gap-dependent and gap-independent bounds, recovering classic regret bounds for CUCB alg for CMAB-T as a special case.

- The authors run experiments in both biased and unbiased settings.

- Overall, I found the paper well-written and generally easy to follow.  The authors included a number of remarks throughout.

**Weaknesses:**

### Major

- My primary concerns are on novelty.  The algorithm H-UCB appears to be CUBC with the paired UCB estimators from Cheung and Lyu (ICML 2024). It is unclear to me how much technical novelty there is in adapting CUCB regret bounds in light of Cheung & Lyu’s prior work on adapting MAB.

   - In Section 4, the authors state “we design a hybrid confidence bound...” that is identical to Cheung and Lyu’s (their Alg 1 step 6) without acknowledgement (at least in that section). (The authors also did not clearly explain the intuition behind that particular formula (e.g. if it came from a variant of the Hoeffding bound for mixed data))

   - The authors in lines 057-066 comment on how there are differences in how the regret is analyzed for UCB in MAB vs CUCB in CMAB-T, then in lines 085-093 have discussion that makes it sound like they had to deeply rethink CMAB-T analysis and come up with new ways of integrating (possibly biased) offline data.  But given how the algorithm design ended up just being classic CUBC with Cheung and Lyu’s UCB estimators, I am skeptical of the extent of novelty claimed.



- I have some concerns about motivation for the specific distributional shift in CMAB-T.  The authors did not provide a concrete example in the set up regarding what the base arm means are in relation to the triggering process and the reward function.  It is consequently unclear to be if it is well-motivated to consider that the learner would have prior knowledge on distributional shift for the base arms (including non-trivial $V_i$’s) but that there would be no shift at all on the triggering process.  Based on just the abstract problem setup, I would expect that shifts in both sources of randomness behind the super-arm rewards would be accounted for in a hybrid CMAB-T problem.



### Minor

- The authors only consider $[0,1]$ base arm rewards, for which the learner can trivially use mean shift bounds of $V_i=1$.  Esp. with the impossibility result of Cheung and Lyu (2024), the results cannot be widely generalized.



- The authors should explicitly mention what classes of reward functions satisfy the assumptions.  The reward model seemingly has a linearity in its expected value (line 163), monotonicity, and a bounded smoothness.  They point out this is standard in the CMAB-T works.

- line 321 the measure $\omega_i$ is not discussed why it has the form it does, such as why it has a directionality wrt the bias – eg even with a conservative (for [0,1] rewards) bound $V_i=1$ for base arm i,

    - if the off and on means are the same (unbiased), $\omega_i=1$ (just equal to our pessimism)

    - if the bound is tight with arm i looking better offline than online, $\omega_i=2$

    - if the bound is tight with arm i looking worse offline than online, $\omega_i=0$

- The experiments are not particularly convincing

   - the authors only compare against purely online and purely offline methods.  The authors cited past works that considered hybrid settings without distributional shift but no baseline was used for that with comparison.  For instance, could one run CLCB and have that warm-start CUCB?

   - The plots (V=0 vs V>0) are confusing to compare. I thought all 6 sub-figures were identical online environments at first.  They must be different, as CUCB performs differently in Fig 1 and 2.   This is concerning as Fig 1 showing with $N=200$ there is a huge gap in the setting without shift

   - The proposed method appears nearly identical with the purely online CUCB baseline even with small bias (V=0.2).



- “Together, these two bounds form a comprehensive characterization of the gap-independent regret in H-CMAB-T” sounds a bit exaggerated – Theorem 2 is the regret upper bound for one algorithm.  A few corner cases are checked, but there is no lower bound included for the H-CMAB-T problem.

- There is no discussion of works on non-stationary bandits. I don’t know if there are any on CMAB-T specifically, and it is of course not the same as having a offline dataset, but there may be some interesting ideas from that setting that would be relevant for this problem.



### Very Minor

- Cheung & Lyu 2024 is cited as an arxiv preprint

**Questions:**

- Regret bound (Thm 1)

    - Does it hold for any $T$? or is there some minimum $T$ (function of problem parameters) it holds for (or for which it becomes non-vacuous)?

   - (minor) Is Alg 1 anytime?  As written it requires the horizon as input, though it is not clear if that is used.





- Does the triggering have any discernible impact on the regret bounds regarding the impact of base arm biases?  eg a measure $\omega_i$ is mentioned but that only is wrt individual arm means.  I don’t know if it is an artifact of the regret analysis of CMAB-T.

- Did you run experiments where you validated if the $\omega$ dependence on $V_i$ and the means in the regret upper-bounds was consistent with the empirical regret from experiments.  That is, if the offline $\mu_i$’s are always better than online $\mu_i$’s and the $V_i$ values are tight, would the algorithm’s performance be identical with Figure 1 where there is no bias (and it is known there is no bias)?

---

> ### Author Response · Authors · 2025-11-26
> **Rebuttal for Reviewer 9DtE part1**
>
> We thank the reviewer 9DtE for providing valuable feedback. Please see our response below.
>
> **Novelty**
>
> We thank the reviewer for pointing this out, and we apologize for not explicitly emphasizing the connection to [1] in Section 4.
> We agree that the hybrid confidence bound we use has a similar structural form to the estimator in [1]’s Algorithm 1 Step 6, and we will update the paper to acknowledge this connection clearly.
>
> However, we would like to stress that this similarity is not an overlap in contribution, but rather a necessary and natural design choice when extending UCB-style methods to hybrid CMAB-T.
> Both CMAB-T and hybrid MAB rely on the same classical principle:
> *Construct a confidence radius that combines online noisy samples and offline biased-but-bounded samples.*
>
> Given this shared statistical setting,
> any high-probability estimator that integrates both sources must contain the “online deviation term + offline bias compensation term” structure, regardless of whether the domain is MAB, CMAB, or CMAB-T.
> Thus, when designing the first hybrid UCB for CMAB-T (not MAB), it is natural to start from:
>
> 1.the standard CMAB-T UCB structure ([2]; [3]), and
>
> 2.the hybrid-MAB confidence bounds ([1]),
>
> and ask whether a unified hybrid-UCB can be analyzed in the CMAB-T setting. Although the confidence bound looks similar in algebraic form,
> the regret analysis in hybrid CMAB-T is fundamentally more difficult and cannot be obtained from hybrid MAB. The analytical and theoretical contributions are listed.
>
> **Analytical techniques.**
> The main challenge in the hybrid CMAB-T setting lies in deciding, at each round, which UCB estimate to use: the one based on biased offline data or the one from online interactions. In the simpler MAB setting studied by [1], this decision is easier because the regret can be directly decomposed using suboptimality gaps $\Delta_i$ and the number of arm pulls. This decomposition enables a relatively straightforward threshold analysis between hybrid UCB estimates and the optimal mean $\mu^*$.
> However, in CMAB-T, such a gap-based regret decomposition does not apply. Instead, regret must be controlled through confidence intervals and triggering probabilities, which do not naturally lend themselves to closed-form thresholds. A naïve approach would be to derive a switching rule by solving the inequality between the two UCB estimates. Unfortunately, this leads to high-degree, analytically intractable expressions that lack interpretability.
>
>
> To address this challenge, we propose a principled decision rule that is tightly coupled with the uncertainty-based structure of CMAB-T analysis. The core idea is to incorporate offline data only when the reduction in uncertainty it brings outweighs the potential error due to bias, that is, when $\omega_i\leq M_i/2BK$. Specifically, we analyze arm-level bias and aggregate it at the super-arm level via the triggering mechanism, and discuss the values of such aggregated bias and the per-round regret. This approach yields a regret analysis that naturally balances robustness and adaptivity, and it reveals how offline data can be selectively leveraged to improve performance without compromising the theoretical guarantees of pure online learning.
>
> **Theoretical results.** Our theoretical results also reflect a fundamental difference from those in [1]. While [1] establishes a regret bound of the form $O({m \log T}/{\Delta} - \sum_i N'_i \Delta_i)$, our bound takes the form $O({m \log T}/{\Delta} - \sum_i \sqrt{N_i' \log T})$. The saving term $- \sum_i N'_i \Delta_i$ in [1] intuitively captures the reduced number of times suboptimal arms are selected. In contrast, our saving term $- \sum_i \sqrt{N_i' \log T}$ reflects the reduction in uncertainty enabled by the presence of offline data—highlighting a distinct interpretation and analytical perspective.
>
> Uncertainty-based analysis plays a central role in many complex learning problems, such as linear bandits and linear MDPs. In contrast, per-arm analysis lacks this adaptability and does not extend naturally to such settings. We believe that our analytical approach, grounded in the trade-off between uncertainty and bias, can be generalized to a broader class of learning problems.

---

> ### Author Response · Authors · 2025-11-26
> **Rebuttal for Reviewer 9DtE part2**
>
> **Discussion on p.**
>
> We agree that the interaction between distributional shift and the triggering process is subtle, and we clarify below why our formulation focuses on shifts in the mean parameters while keeping the triggering mechanism fixed.
>
> (1) In many concrete CMAB-T models, the triggering probabilities are functions of the base-arm means. This is especially true in practical and widely studied CMAB-T instances:
>
> • **Cascading / LTR (Learning-to-Rank)**: In LTR, the environment first samples a Bernoulli click outcome for every item according to its mean $\mu_i$.
> The triggering process is then:
>
> item 1 is always triggered;
>
> item 2 is triggered iff item 1’s click sample is 0;
>
> item 3 is triggered iff item 1 and 2 are both 0; etc.
>
> Hence the triggering probabilities satisfy $p_{S,i}=\Pi_{j<i}(1-\mu_j)$, which depend directly on $\mu_j$.
>
> Therefore, a shift in means already induces a shift in the triggering process.
> This is why in our hybrid setting, a bias on the means implicitly induces a bias on triggering probabilities—no separate modeling is needed.
>
> • **OIM (Online Influence Maximization)**: Triggering is governed by independent edge activations whose success probabilities are exactly the base-arm means.
> Again, any shift in the means directly changes the triggering process.
>
> Thus, for the most canonical CMAB-T applications, shifting means is equivalent to shifting triggering probabilities.
>
>
> (2) If desired, we will add a dedicated appendix section showing explicitly, for LTR and OIM:
>
> • how mean shift $\mu_i$ to $\mu_i+V_i$ induces changes in $p_{S,i}$,
>
> • how these induced shifts are bounded by the same bias constraint $V_i$,
>
> • and how this interacts with our hybrid confidence bound.
>
> This will make the motivation completely transparent.
>
> ----
>
> **Assumptions on reward functions**
>
> To clarify the reward assumptions, we provide a concrete example using the Learning-to-Rank (LTR) / cascade bandit model, one of the most widely studied CMAB-T instances.
>
> In the cascade model, for a list
> $S=(i_1,...,i_K)$,
> let each item $i$ have a click mean $\mu_i$.
> The user examines the list from top to bottom, and clicks the first attractive item.
> The expected click probability of the list is
> $r_S(\mu)=1-\Pi_{i\in S_t}(1-\mu_i)$.
> which can easily check that it satisfies monotonicity and TPM.
>
> Notably, we do not make the assumptions that the reward function is linear, and in fact any linear reward function can satisfy TPM.
>
> ----
>
> **Discussion on $\omega_i$**
>
> We clarify the design and interpretation of the quantity
> $\omega_i = \mu_i^{\mathrm{off}} - \mu_i^{\mathrm{on}} + V_i$, and why it
> captures the directionality of the offline bias.
>
> **(1) Purpose of $\omega_i$.**
> The quantity $\omega_i$ combines two inseparable pieces of information:
> (i) the actual offline bias $(\mu_i^{\mathrm{off}} - \mu_i^{\mathrm{on}})$,
> and (ii) the pessimism window $V_i$ that specifies the uncertainty level of the
> offline data.
> This determines how much the algorithm can trust the offline samples of arm $i$.
> In our opinion, a regret bound is satisfactory only when it incorporates both
> the hyperparameter $V_i$ and the actual bias $(\mu_i^{\mathrm{off}} - \mu_i^{\mathrm{on}})$.
>
>
> **(2) Case that $\omega_i = 0$.**
> When $\mu_i^{\mathrm{off}} - \mu_i^{\mathrm{on}} = -V_i$, the bias is fully
> compensated by the pessimism bound, yielding $\omega_i = 0$.
> In this case, the offline data makes the suboptimal arm look worse than the
> true online mean, which *accelerates* elimination.
> Therefore using such offline data does not hurt the algorithm.
>
> **(3) Case that $\omega_i = 2$.**
> When $\mu_i^{\mathrm{off}} - \mu_i^{\mathrm{on}} = +V_i$, the offline data
> makes the suboptimal arm look better by the maximum allowed amount.
> This is the hardest scenario for elimination, and the regret bound correctly
> reflects this by producing $\omega_i = 2$.
>
> **(4) Example.**
> In an extreme but valid scenario, if all suboptimal arms have offline means
> significantly underestimated (e.g., $\mu_i^{\mathrm{off}} \approx 0$), the
> algorithm can identify the best arm almost instantly. This matches our bound
> when $\omega_i$ is small.

---

> ### Author Response · Authors · 2025-11-26
> **Rebuttal for Reviewer 9DtE part3**
>
> **Evaluation**
>
> **(1) Baselines:  warm-start CUCB with offline data with bias**
>
> Indeed, warm-starting CUCB using offline CLCB estimates is a natural hybrid baseline and has been used in prior work without distributional shift.
> Our focus in this paper is on the more challenging setting with *biased* offline data, where directly warm-starting CUCB using possibly-shifted offline estimates can introduce additional bias.
> Nevertheless, we agree that including this baseline would strengthen the empirical section.
> We will incorporate a warm-start CUCB baseline (which in fact is the H-CUCB but always setting the input $V=0$) in the future version.
> Our algorithmic design is fully compatible with this baseline, and we expect its performance to be worse than CUCB and our proposed hybrid method.
>
> **(2) CUCB performs differently in plots for $V=0$ vs. $V>0$**
>
> The reviewer is correct: in the current implementation, the online environment was inadvertently re-sampled for different values of $V$, which results in slightly different CUCB curves in Fig. 1 and Fig. 2.
> This was an implementation oversight rather than an algorithmic issue. This is because the online environment is built after checking whether the bias is zero. As a result, when $V>0$ the CUCB performs uniformly despite different $V_i$ but differently between $V=0$ and $V>0$.
>
> In the future version, we will fix the online environment across biased and unbiased cases.
>
> **(3) The proposed hybrid method appears almost identical to pure CUCB when the bias is small.**
>
> We would like to clarify that a bias level of $V=0.2$ is in fact already substantial: it corresponds to a $20\%$ deviation in reward probability, which can be comparable to or even larger than the inherent gaps between optimal and suboptimal arms in many bandit/LTR environments.
> In such cases, even a moderately biased offline dataset provides only limited reliable information for initialization or guidance, and a cautious algorithm will naturally behave similarly to a purely online method.
> If the arm gaps were significantly larger than the bias, the online learning problem itself would be trivial; hence this regime is realistic and nontrivial.
>
> Furthermore, this phenomenon is consistent with what has been observed in prior hybrid MAB experiments in [1]: once the offline bias exceeds a moderate threshold (e.g., $V>0.2$), hybrid algorithms often revert to behavior nearly indistinguishable from pure online UCB.
>
> **(4) Comparison between $w_i=0$ but $V_i\neq 0$ and unbiased case**
>
> From the theoretical side, our regret analysis shows that when $w_i = 0$ for all arms, the bias does not propagate into the hybrid estimator, and the regret upper bound reduces exactly to the unbiased case, even if $V_i \neq 0$ numerically.
> We have not included explicit experiments for this special setting in the current submission, but we agree that it provides a meaningful sanity check connecting the empirical behavior with the structure of the regret bound and will add a dedicated experiment in the future version where we set $w_i = 0$ while allowing $V_i \neq 0$.
>
> ----
>
> **Lower Bound**
>
> In the revised PDF, we have added a full and rigorous proof in Appendix E.
> Here we summarize the key idea and construction.
>
> We consider a CMAB-T gating instance with triggering probabilities
> $p_{S,i} = 1/K$ and linear expected reward
> $r_S(\mu) = \frac{B}{K} \sum_{i \in S} \mu_i$.
>
> For each suboptimal arm $i$, we construct a neighboring environment $\nu^{(i)}$
> in which only $\mu_i$ is increased by $\Delta$, while keeping the offline
> distributions as close as possible so that the algorithm cannot easily
> distinguish the two.
>
> By a standard change-of-measure argument, the online and offline observations
> must satisfy
> $$
> \mathbb{E} _ {\mu}[T_i]\Theta(\Delta^2)+N_i(\Delta - 2V_i) _ {+}^{2} \ge\log T .
> $$
>
> Solving for $\mathbb{E} _ {\mu}[T_i]$ and substituting it into the regret identity
> $\mathrm{Reg} _ i(T)= B\Delta_i\mathbb{E}_{\nu}[T_i]$ gives
> $$
> \mathrm{Reg}_i(T) \ge B\left(\frac{\log T}{\Delta_i}-\sqrt{N_i''\log T}\right),
> \qquad
> \text{with $\Delta_i$ chosen optimally.}
> $$
>
> Summing over all suboptimal arms yields a lower bound with the same structure
> as our upper bound.  All intermediate derivations are provided in the revised
> Appendix E.

---

> ### Author Response · Authors · 2025-11-26
> **Rebuttal for Reviewer 9DtE part4**
>
> ----
>
> **Discussion of works on non-stationary bandits**
>
> We acknowledge that, during the development of this work, we did not fully explore the potential conceptual connections with the literature on non-stationary bandits.
> While the technical settings differ substantially—non-stationary bandits model reward distributions that evolve over time, whereas our problem concerns a fixed offline distribution that may differ from the online one—the underlying issue of how much trust to place in past observations is indeed related in spirit.
>
> We appreciate the reviewer for pointing out this overlooked connection.
> In the future version, we will add a discussion of relevant non-stationary bandit techniques such as discounting, sliding-window estimation, and change-point detection, and how these ideas might offer inspiration for designing more adaptive hybrid algorithms under distributional shift.
> We view this as a valuable direction for future work, and we thank the reviewer for bringing it to our attention.
>
> ----
>
> **The T in Theorem 1 and algorithm 1**
>
> 1.On Theorem 1:
>     The bound we present is a standard gap-dependent regret upper bound of the form commonly used in stochastic bandits.
>     Such bounds hold for *any horizon $T$*; they do not require a minimum $T$ to become non-vacuous.
>     When $T$ is small, the confidence level $\delta$ can be chosen accordingly (as in classical analyses), and the resulting bound scales smoothly with $T$ without requiring asymptotic assumptions.
>
> 2.On Algorithm 1:
>     Although Algorithm 1 is written with $T$ as an input, the horizon is not used in the update rule itself.
>     This is typical in bandit algorithms where $T$ is included only for convenience in the analysis.
>     The algorithm can be made fully *anytime* by either (i) removing $T$ entirely from the pseudocode, or (ii) applying the standard doubling-trick argument, which does not affect the regret guarantees.
>
> ----
>
> **Does triggering have any discernible impact on how base-arm biases affect regret?**
>
> First of all, in CMAB-T, the triggering probability of each base arm is a deterministic function of its mean reward.
> Thus the discrepancy between offline and online reward distributions---captured by $w_i$---already encodes how often an arm is expected to be triggered.
> In other words, the quantity $w_i$ includes the effect of triggering because triggering is itself governed by the underlying means.
>
> From the technical perspective, our analysis follows the standard Triggering Probability Estimation (TPE) technique of [4], which converts random probabilistic triggering into the expected membership of each arm in the triggering set.
> Once this transformation is applied, the problem can be analyzed entirely at the ``per-arm’’ level: each arm contributes to regret proportionally to its expected presence in the triggering set, and the confidence bounds on individual arms determine the overall regret.
> And most importantly, even in the hybrid setting, the triggering structure of the *online* environment remains unchanged.
> Offline data influence only the decision rule, not the triggering process itself.
> Therefore, after applying TPE, the offline bias affects regret through $w_i$, while the impact of triggering is naturally absorbed into the expectation of the triggering set as in the standard CMAB-T formulation.
>
> For these reasons, the appearance of $w_i$ is not a proof artifact: it is the correct per-arm quantity once triggering has been reduced to its expected effect via TPE.
> This connection is also clarified in the detailed proof in Appendix B, roughly from line 694 to line 736.
>
> ----
>
>
> *[1]Wang Chi Cheung and Lixing Lyu. Leveraging (biased) information: Multi-armed bandits with
> offline data. arXiv preprint arXiv:2405.02594, 2024. doi: 10.48550/arXiv.2405.02594. Accepted
> to ICML 2024.*
>
> *[2]Wei Chen, Yajun Wang, Yang Yuan, and Qinshi Wang. Combinatorial multi-armed bandit and its
> extension to probabilistically triggered arms. Journal of Machine Learning Research, 17(50):1–33,
> 2016.*
>
> *[3]Wang Chi Cheung and Lixing Lyu. Leveraging (biased) information: Multi-armed bandits with
> offline data. arXiv preprint arXiv:2405.02594, 2024. doi: 10.48550/arXiv.2405.02594. Accepted
> to ICML 2024.*
>
> *[4]Xutong Liu, Jinhang Zuo, Siwei Wang, John C.S. Lui, Mohammad Hajiesmaili, Adam Wierman, and
> Wei Chen. Contextual combinatorial bandits with probabilistically triggered arms. In Andreas
> Krause, Emma Brunskill, Kyunghyun Cho, Barbara Engelhardt, Sivan Sabato, and Jonathan
> Scarlett (eds.), Proceedings of the 40th International Conference on Machine Learning, volume
> 202 of Proceedings of Machine Learning Research, pp. 22559–22593. PMLR, 23–29 Jul 2023.*

---

### Note · Authors · 2026-01-13

**Comment:**

We would like to withdraw this submission in order to submit the work to another venue (TMLR).
Thank you for your time and consideration.

**Withdrawal Confirmation:**

I have read and agree with the venue's withdrawal policy on behalf of myself and my co-authors.